# Targeting Epigenetic Dependencies in Solid Tumors: Evolutionary Landscape Beyond Germ Layers Origin

**DOI:** 10.3390/cancers12030682

**Published:** 2020-03-13

**Authors:** Francesca Citron, Linda Fabris

**Affiliations:** 1Department of Genomic Medicine, The University of Texas MD Anderson Cancer Center, Houston, TX 77054, USA; fcitron@mdanderson.org; 2Department of Experimental Therapeutics, The University of Texas MD Anderson Cancer Center, Houston, TX 77054, USA

**Keywords:** epigenetics, cancer stem cell, targeted therapy, ovarian cancer, glioblastoma, pancreatic cancer

## Abstract

Extensive efforts recently witnessed the complexity of cancer biology; however, molecular medicine still lacks the ability to elucidate hidden mechanisms for the maintenance of specific subclasses of rare tumors characterized by the silent onset and a poor prognosis (e.g., ovarian cancer, pancreatic cancer, and glioblastoma). Recent mutational fingerprints of human cancers highlighted genomic alteration occurring on epigenetic modulators. In this scenario, the epigenome dependency of cancer orchestrates a broad range of cellular processes critical for tumorigenesis and tumor progression, possibly mediating escaping mechanisms leading to drug resistance. Indeed, in this review, we discuss the pivotal role of chromatin remodeling in shaping the tumor architecture and modulating tumor fitness in a microenvironment-dependent context. We will also present recent advances in the epigenome targeting, posing a particular emphasis on how this knowledge could be translated into a feasible therapeutic approach to individualize clinical settings and improve patient outcomes.

## 1. Introduction

Epigenetics is a concept first introduced in the field of developmental biology in 1942 when Conrad Waddington defined it as *“the causal interactions between genes and their products, which bring the phenotype into being”* [1]. This definition has evolved over time, and, nowadays, the concept has been extended to a wide range of cellular processes and cell states determined by hereditary chromatin structure and not directly determined by alterations of DNA sequence [2]. Thus, the study of chromatin architecture and spatial distribution, exactly as the DNA sequence, is posed in the center of evolution, profoundly altering gene expression patterns through the compactness and accessibility of DNA [3,4]. In this context, chromatin is the macromolecular complex of DNA and histone proteins, which provides the scaffold for the efficient packaging of 3 billion pair length of human DNA into a ~10-µm cell nucleus [2,5]. 

The nucleosome is the fundamental and repeated unit which constitutes the basic functional structure of chromatin. A nucleosome is formed by a ~150-bp double-stranded DNA wrapped around and constituted by two molecules of each histone H2A, H2B, H3, and H4. Chromatin compactness is further ensured from the histone linker H1 and other cofactors such as methyl-CpG-binding protein 2 (MeCP2), heterochromatin protein 1 (HP1), high mobility group (HMG) proteins, poly(ADP-ribose) polymerase 1 (PARP1), myeloid and erythroid nuclear termination stage-specific protein (MENT), and Polycomb group proteins, which lead to cohesive chromatin structures [3,5].

The local and global chromatin architecture is profoundly influenced by dynamic epigenetic modifications, and it is inherited from daughter cells across several cycles of divisions. Thus, it is not surprising that the complexity of the “epigenetic landscape” directly lies at the heart of cell evolution and profoundly contributes to delineate cell identity beyond the DNA sequence, implicating sophisticated regulations of self-renewal and lineage commitment [2,6,7].

In this scenario, the epigenetics revolution stands in the way of the long-held traditional dogma of biology where genetic information has a unidirectional flow, and the coding genes are the single determinant entities of cell morphology and function. Considering that each single cell of our organism has the same genome, the extent of morphologically and functionally different cells, sitting within the same organ, is impressive. The question should be, how does this occur? Probably we still lack the ability to draw a complete portrait of the cell complexity, but the interplay between the genomic and epigenomic heritages significantly contributes to the phenotype complexity at the base of cell evolution. 

Chromatin is a dynamic model of the genome, which continuously and incessantly receives and elaborates signaling to remodel and re-adapt the gene expression in response to both cell-dependent and -independent mechanisms. 

Starting from embryogenesis, DNA and chromatin are responsible for the hierarchical development of any human tissue, driving the transition from multipotent stem and progenitor cells to terminally differentiated and specialized progenies [2,8,9].

Dynamics of gene expression pattern is mediated by epigenetic modifications, which include DNA methylation/demethylation, covalent histone modification, and non-covalent mechanisms such as RNA interference [5,6,7,9,10,11,12,13,14]. 

In this review, we will give an overview of the principal epigenetic dependencies of solid tumors, starting from the molecular mechanisms of epigenetic modifications and the consequences of their unbalanced workflow to the pharmacological approach targeting this complex pathway, with the final goal of improving patient outcomes.

## 2. Molecular Mechanism of Epigenetic Modifications

Epigenetic alterations, in concert with other genetic changes, are responsible for genomics, transcriptomics, and proteomics alterations. Epigenetic changes involve DNA methylation, histone modifiers and readers, chromatin remodelers, microRNAs, and other components of chromatin. Epigenetic states are flexible, and they mainly occur through three different forms: methylation of DNA, post-translational modification of histones, and through organization of chromatin structure.

The best-characterized epigenetic modification is DNA methylation, which occurs on the 5-Carbon on cytosine residues of the CpG dinucleotides. These nucleotides are specifically localized in GpC high-content stretches defined as CpG islands, usually found in the proximity of gene regulatory regions (either promoters or enhancers) [15]. The methylation reaction is catalyzed by a group of enzymes, the DNA methyltransferares (DNMTs), also defined as “Writers”. There are three different isoforms of DNMT enzymes that show different and non-redundant roles: DNTM1 is critical to maintaining the proper DNA methylation during mitosis, whereas DNMT3A and DNMT3B are more implicated in the methylation process during embryogenesis [16]. Globally, the readout of the DNMTs activity is the hypermethylation of DNA, which is usually associated with the silencing of gene expression [17,18,19]. Aberrant DNMT levels has been detected in many different cancer types through genome-wide analyses, which highlighted a ~5–10% increased methylation on CpG islands in cancer specimens compared to normal counterparts, with a consequent silencing of several tumor-suppressor genes (i.e., CDKN2B, RASSF1A, GATA4, CDH1), further supporting the tight correlation between epigenetics abnormalities and cancer initiation and progression [20,21,22,23,24].

On the opposite side, there are TET1 and TET2, the ten-eleven translocation enzymes, methyl-cytosine dioxygenases known as the DNA methylation “Erasers”, responsible for catalytically converting 5mC to hydroxy-methyl-cytosine (5hmC) [19,25].

The histone modeling encompasses a broad range of molecular covalent modifications occurring in the N-Terminal of a lysine (K) residue of histones. Methylation and acetylation/deacetylation are the most prevalent in terms of frequency, thus are well characterized, but ubiquitylation, sumoylation, phosphorylation, biotinylation, and ADP-ribosylation reactions are also contributing to the dynamic regulation of gene expression [23,26,27,28,29]. Depending on the modification reaction and the lysine where it occurs, the final result could be either gene expression activation or repression due to an “open” or “close” architecture of the chromatin, also called heterochromatin or euchromatin, respectively. Heterochromatin is highly condensed, late to replicate, and primarily contains inactive genes, while euchromatin is relatively open and contains most of the active genes.

Lysine acetylation is mediated by histone acetyltransferases enzymes (HATs) and is associated with open chromatin and a transcriptionally permissive state. On the other hand, deacetylation is regulated by the histone deacetylases (HDACs) and converts chromatin to a more condensed or transcriptionally repressed state [23,24]. 

Based on the function, molecular mass, and pairwise sequence similarity with yeast counterpart, the HDAC family consists of 18 members divided into two families and classified into four categories. The two families refer to classical HDACs (i.e., Classes I, II, and IV) and non-canonical Sirtuins (i.e., Class III). 

HDACs 1, 2, 3, and 8 belong to the canonical class I, and they are mainly involved in the regulation of cell proliferation and apoptosis. Class IIa (HDACs 4, 5, 7, 9) and Class IIb (HDACs 6 and 10) display a large NH2-terminal domain that regulates HDACs nucleus/cytoplasm shuttering and DNA binding. Their expression appears to be strictly regulated, in agreement with their role in cell differentiation and development. Class IV accounts only for HDAC 11 [30].

The canonical Classes I, II and IV are characterized by a zinc-finger domain; thus, a zinc ion in the enzymatic pocket is necessary for the hydrolase activity. Indeed, all zinc-dependent HDACs can be inhibited by zinc-chelating agents such as hydroxamic acid. The non-canonical Class III HDACs displays high similarity with yeast Sirtuin protein (SIR2) and also shows NAD+-dependent ADP-ribosyltransferase activity [31]. These enzymes also differ for cellular localization and expression: Class I HDACs are localized only in the nucleus and are the most abundant, whereas Classes II, III, and IV can be found both in the nucleus and in the cytoplasm [32]. An example of the non-epigenetic function of these enzymes is represented by the increasing number of non-histone proteins recognized as HDAC substrates (i.e., p53, E2Fs, GATA1, Bcl-6, STAT3, HMGs, HSP90, NF-κB, tubulin, importin, nuclear hormone receptors, and β-catenin). 

For instance, the major cytoplasmic deacetylase HDAC6 is involved in the regulation of cell motility, adhesion, and chaperone through the deacetylation of α-tubulin and HSP90 [30,33]. Similarly to DNA methylation, aberrant HDACs expression has been observed in different cancer types, and it is linked to a general histone hypoacetylation leading to a transcriptionally active chromatin state [34].

Another important histone modification is the histones H3 and H4 methylation, mediated by histone methyltransferase enzymes (HMTs). Depending on the methylated residues, these enzymes are classified into lysine methyltransferases (KMTs) or protein arginine methyltransferases (PRMTs). Histone methylation has a double role; it can function as a repressor or activator on the basis of which residue is modified and the degree of methylation (i.e., lysine 4 trimethylation on histone H3—H3K4me3-, H3K36me3 and H3K79me3—indicates a transcriptionally active region, but H3K9me3 or H3K27me3 have negative regulatory effects on gene promoters) [5,7].

Furthermore, bivalent or “poised marks”, containing both repressive (H3K27me3) and active (H3K4me3) histone modifications, appear to identify pluripotency genes.

Finally, RNA interference can also alter gene expression by forming antisense transcripts, which prevent the accumulation of homologous transcripts. For example, non-coding RNA such as microRNAs, small RNA molecules of ~22 bp, display a pleiotropic activity by canonically binding thousands of complementary mRNA transcripts and, in turn, perturbing their stability and subsequent translation [35,36]. Other involved mechanisms are the long non-coding RNAs (lnRNAs) that may serve as adaptors to drive epigenetic modulators or may sequestrate epigenetic complexes or transcription factors away from the chromatin [37,38] (Figure 1). 

## 3. Epigenetic Reprogramming: the Common Switch Through Tumor De-Differentiation

De-differentiation is recognized as the morphological loss of lineage identity, and it is associated with high tumor heterogeneity and poor prognosis in different cancer types [39,40]. However, how cell de-differentiation is linked to both tumorigenesis and cancer development is still unclear and undoubtedly represents an emerging field of study [41,42,43]. 

Recent pathological and molecular studies have nicely suggested a tight relationship between de-differentiation, cancer stem cells (CSCs), and tumor heterogeneity [44,45,46]. Indeed, de-differentiation is thought to be driven by the reprogramming of somatic cells toward the acquisition of unlimited proliferation properties and self-renewing activities, which are typical characteristics of cancer cells, and originally of stem cells. Moreover, de-differentiated cells display a dynamic phenotypic plasticity, endowing their adaption in response to environmental perturbations (i.e., nutrient deprivation, oxidative stress, inflammation, treatment-related selective pressure), shuttling from differentiate phenotype to an undifferentiated one and vice versa, further contributing to tumor heterogeneity [2,46,47].

This is also supported by common physiological and molecular traits found between embryonic stem cells and CSC, and these close similarities were observed at gene expression and epigenetic levels [43,48]. 

Specifically, CSCs are characterized by an “open” chromatin structure, high expression of epigenetic modulators, and frequent histone bivalent marks. In a simplified scenario, this bivalent chromatin, usually found in embryonic stem cells, maintains the developmental genes in a poised state for later activation or repression in a time and context-specific manner. Moreover, CSCs re-activate the so-called embryonic signaling pathways, including Wnt/β-catenin, Hedgehog and Notch, but also IL6/JAK/STAT3, NFκB, and PI3K/AKT, which in turn control the epigenetic dynamics in the cells in a positive feedback loop [49,50,51,52,53,54]. 

Looking deeply into the cell biology, the final readout of a cell’s identity lies in its unique gene expression, which delineates its variable property or plasticity of responsiveness to different environmental, developmental, or metabolic cues. It is conceivable that discerning the regulatory network beyond context-specific gene expression associated with a myriad of cell states falls back in a long-standing scientific goal with immediate clinical implications.

From a clinical standpoint, the targeting of cancer epigenome inspires the idea of tumor reprogramming. The generation of "epigdrugs" that target epigenetic erasers, writers, and readers, drawing new chromatin architectures in a dynamic fashion, have already successfully endowed the manipulation of cells and microenvironment to reshape cell fate in stem-cell based therapy and regenerative medicine [55,56]. 

From a translational viewpoint, the innovative reprogramming of epigenetic machinery could be a powerful tool towards personalized, dynamic, and reversible therapy, finally representing a reliable opportunity to refine cancer patients’ selection and to improve their outcomes, avoiding unnecessary toxicities.

## 4. Epigenetic Modulators: Hitting the Leitmotiv across Different Germ Layers-Derived Tumors

There is still poor knowledge about the entities involved in tumor initiation, maintenance, and eventually progression after the treatments. Indeed, it not surprising that CSCs, characterized by poised chromatin, phenotypic plasticity, and a relative quiescent state, are thought to be the major reservoir for tumor sustaining, not only in solid tumors [9,47].

In this review, we will specifically focus on the CSCs epigenetic vulnerabilities across tumors arising from the three different germ layers: glioblastoma multiforme (GBM), high grade epithelial ovarian cancer (HGEOC), and pancreatic ductal adenocarcinoma (PDAC), respectively originating from the ectoderm, mesoderm, and endoderm.

These tumors are characterized by the silent onset, causing a delayed diagnosis (i.e., Stage III–IV), and acquired or intrinsic resistance to therapy [57,58]. In this scenario, loss of asymmetric cell division, clinically displayed in late-stage tumors, can potentially be considered a critical common feature between these cancer types. The promotion of symmetric self-renewal towards an undifferentiated phenotype may have direct implications in the chemotherapy-induced escape strategy, which could be, lately, inherited by CSC offspring [59,60].

GBM, according to WHO, is classified as Grade IV astrocytoma. It is the most frequent of all brain tumors, accounting for 60% of all the cases in adults. The median survival time is about one year, and it dramatically drops to 2–3% when the 2-year follow-up is considered. 

The histopathology of GBM resembles the one of anaplastic astrocytoma, with pleomorphic cells ranging from small and poorly differentiated cells to large multinucleate cells, high mitotic activity, and multifocal necrosis [61]. Based on molecular pathogenesis and gene expression profiles, GBM is classified in classical, mesenchymal, proneural, and neural subtypes [62,63]. Despite the molecular heterogeneity, GBM patients are treated with first-line temozolamide (TMZ), which remains only an elusive clinical option [62,64].

HGEOC represents more than 75% of all EOC and encompasses serous, endometriod, and undifferentiated tumors. All HGEOCs virtually display inactivation of the TP53 gene due to somatic mutations and a high degree of genomic instability due to defects in pathways contributing to DNA repair [65].

Despite the different classification, HGEOC patients are currently treated with first-line platinum-based chemotherapy plus a taxane, independently from their histotypes. In this context, less than 30% of HGEOC patients survive more than five years [66]. Major causes are identified in diagnosis at advanced stages (III–IV) when the disease has already spread in the abdomen and has acquired or intrinsic resistance to therapy, features that have been linked to CSC and cell plasticity [59,67].

PDAC represents approximately 90% of all pancreatic tumors. It represents a rare malignancy (~3% of all cancers), but it is the fourth leading cause of cancer-related death in Western countries, with only 25% overall survival one year after diagnosis, and less than 10% after five years, being tremendously dependent on the stage at diagnosis [68]. Large-scale genomic sequencing revealed mutations in KRAS as a nearly universal event, playing a pivotal role in PDAC cell plasticity, triggering the reprogramming of acinar cells into duct-like lineages capable of progress towards an invasive disease [69].

The standard of care for PDAC patients is limited to cytotoxic chemotherapy, primarily FOLFIRINOX and gemcitabine-based regimens. In this context, the diagnosis in advanced or metastatic stages poses a clinical challenge as there are no effective therapeutics to achieve long-term remissions of the disease [70].

### 4.1. DNA Methylation: Implication and Inhibition

As mentioned above, DNA methylation is probably the most studied epigenetic modification in cancer. Independently from the specific tissue of origin, tumors are broadly characterized by DNA hypomethylation, occurring on intergenic regions, DNA repetitive and regulatory sequences, which contributes to genomic instability and, less frequently, to activation of oncogenes. On the other hand, hypermethylation of CpG-islands in the proximity of promoter regions frequently leads to the silencing of tumor suppressor genes and has been shown to be a critical hallmark of cancer [13,22,71,72]. 

The DNA methylation status may be interpreted as a tumor fingerprint, resembling the traits of the cell of origin. This could be exploited as a useful clinical tool to (i) identify metastasis of uncertain primary tumor, (ii) to refine the molecular classification of tumors, or (iii) to select patients who could benefit from a specific targeted therapy [64,67,73].

One of the most striking examples arises from gliomas, in which epigenetic profiling has been successfully applied in the clinical routine. In these aggressive brain tumors, mutations occurring on IDH1 or IDH2 metabolic genes are linked to a DNA hypermethylation signature. This novel driver machinery in glioma tumorigenesis allowed the identification of a specific glioma subclass, namely glioma CpG island methylator phenotype (G-CIMP), as reflected in the WHO 2016 classification [74]. Overall, G-CIMP patients, regardless of the tumor grade, display plenty of silenced genes affecting multiple hallmarks of cancer [75].

Another powerful example of the importance of epigenetic in this setting is the methylation on MGMT (O-6-methylguanine-DNA methyltransferase) gene. This gene is crucial for genome stability by preventing mismatch and errors during DNA replication and transcription, and its silencing is associated with lower efficiency in DNA repair after TMZ-induced DNA damage. Indeed, glioma patients with MGMT methylation showed a better clinical response after TMZ administration and longer survival benefit [76].

Aberrant DNA methylation state has also been associated with both tumorigenesis, tumor progression, and dedifferentiation. In this scenario, an interesting example came from pancreatic cancer, in which McCleary-Wheeler and colleagues demonstrated that early pancreatic lesions, known as PanIN-1A, already display aberrant methylation, which could eventually enhance tumor progression [77]. This finding is also supported by different studies on the Mucin protein family, whose expression is critically downmodulated in precancerous lesions with respect to the normal counterpart, and it is paralleled by a promoter hypermethylation of the genes (MUC1, MUC2, MUC4, MUC5AC) [78,79,80,81].

Epigenetic analyses of ovarian cancers and normal tissues revealed an extensive hypermethylation in the proximity of different oncosuppressor promoters. The two top genes are *BRCA1* (Breast Cancer Type 1 susceptibility protein) and *RASSF1A* (Ras association domain-containing protein 1), respectively methylated in 24% and 50% of the HGEOC analyzed. Functionally, BRCA1 is involved in the DNA damage pathway, whereas RASSF1A plays a role in the stabilization of microtubules dynamics. BRCA1 silencing either through mutations or epigenetic mechanisms has direct clinical relevance since the so-called BRCA-ness phenotype predicts a better response to *PARP-1* (poly-(ADP-ribose)-polymerase) inhibitors combined with chemotherapy, in term of progression-free survival [82,83]. Epigenetic silencing through promoter hypermethylation also occurs in *CDKN2A* (cyclin-dependent kinase inhibitor 2A), *DAPK1* (death-associated protein kinase 1, a positive mediator of γ-interferon-induced programmed cell death), and *ICAM-I* (intracellular adhesion molecule 1, known as CD54), although the percentage is less frequent than BRCA1 and RASSF1A (i.e., <10%). The global readout of the epigenetic silencing of these oncosuppressors is the elusion from cell cycle inhibition and activation of apoptotic pathways, leading to increased cancer cells proliferation [13,84,85]. 

Inactivation of CDKN2A or RASSF1A in both PDAC and GBM via promoter methylation appeared to be a common molecular event, and possibly an early event during tumor evolution, as witnessed by the loss of CDKN2A in approximately 95% of PDAC [86,87].

An independent study showed a critical correlation between the promoter methylation of eight tumor suppressor genes (*CDKN2A*; *RASSF1A*; *RARB*—encoding the retinoic acid receptor β; *CDH1*—encoding E-cadherin cell adhesion protein; *CDH13*—known as T-cadherin, which has a unique role in cell migration; *APC*—adenomatous polyposis coli, which is the main repressor of β-Catenin activity; *GSTP1*—glutathione S-transferase P is an important player in the detoxification of carcinogens and cytotoxic drugs by glutathione conjugation; and *MGMT*) and the invasiveness of ovarian cancers, supporting a pivotal role of DNA methylation not only in cell proliferation but also in migration, invasion, and metastasis. 

E-cadherin, encoded by the CDH1 gene, is a key player against EMT (epithelial-to-mesenchymal transition), and it appeared to also be epigenetically regulated in pancreatic cancer. In this case, FOXA1 and FOXA2 transcription factors are downmodulated due to promoter hypermethylation, thus leading to a negative regulation of E-cadherin expression, which is sufficient to stimulate an EMT program and tumor progression [88]. 

Overall, these findings may support a rationale for the clinical application of DNA methylation inhibitors (DNMTi; Figure 1). The first-generation DNMTis is represented by 5-azacytidide and decitabine, which are nucleoside analogues of cytidine incorporated during S-phase and able to intercalate into DNA strands and selectively bind the DNMT, catching it after DNA replication. Moreover, 5-azacytidide is markedly incorporated into RNA, likely altering RNA translation and protein metabolism. 

Preclinical studies demonstrated that 5-azacytidine strongly sensitizes pancreatic cancer cells to radiotherapy; unfortunately, in vivo experiments brought out a range of side effects due to its relative non-specific mechanism of action, inducing a broad hypomethylation. In addition, 5-azacytidine showed typical toxicity of intercalating agents, which prompted the development of a new-class of DNMTis [89].

One of the successfully examples of DNMTi application in the clinic is the treatment of platinum-resistant ovarian cancer patients. Abnormal DNA methylation, as described above, heavily triggers platinum-resistance, and the administration of 5 days of decitabine followed by carboplatin, in platinum-refractory patients, strongly reverted these epigenetics changes [90]. This result correlated with a diffuse hypomethylation, activating the re-expression of tumor suppressor genes, such as MLH1, RASSF1A, HOXA10, and HOXA11, all belonging to DNA repair and immune-response pathways. Decitabine is also able to affect the expression of genes involved in developmental pathways, such as Hedgehog and TGF-β signaling, considered partially responsible for platinum re-sensitization [90,91,92]. Accordingly, decitabine treatment showed a 35% improved response in platinum-refractory patients respect to 5-azacytidine, in term of demethylation index and progression-free survival [90].

The results from a Phase I clinical trial were recently published, demonstrating that the administration of the second-generation DNMTi guadecitabine for 5 days followed by carboplatin was well tolerated and induced a clinical response in heavily pretreated ovarian cancer patients, supporting the beginning of a Phase II study [93].

A controversial scenario came from IDH mutated glioma patients. Treatment with first-generation DNMTis, 5-axacytidine, and decitabine showed interesting results in preclinical in vitro and in vivo models, but completely failed to reach significant clinical activity [94]. In particular, for GBM patients with methylated MGMT, it is still a subject of study whether a general demethylation could be an effective strategy, since it could induce the demethylation of repair gene MGMT, with the potential development of resistance to alkylating agents like TMZ.

Moreover, Pacaud and colleagues reported that the disruption of the DNMT1/UHRF1/PCNA complex, essential for the proper DNA methylation in normal cells, may act as an oncogenic event promoting gliomagenesis [95].

Concordant results came from the preclinical testing of Zebularine, a cytidine-analog DNMTi, in HGSOC, PDAC, and GBM. This drug could be orally administrated and demonstrated to re-sensitize ovarian cancer cells to platinum-based therapy, to elicit the abrogation of PDAC CSCs and to selectively kill GBM cells deficient for DNAPK (DNA-dependent protein kinase involved in the DNA damage pathway) [96,97,98].

The therapeutic activity of these DNMTis is not solely restricted to cell-autonomous mechanisms. In this context, the immunogenic tumor-associated antigen, NY-ESO-1, is broadly silenced through promoter methylation in many cancer types via sophisticated epigenetic orchestration [99]. Indeed, guadecitabine is sufficient to restore the expression of NY-ESO-1, potentially stimulating the innate immune response when combined with Class I and Class II HDAC inhibitors [100]. This mechanism could encourage the use of immune checkpoint blockade in tumors such as GBM, and HGSOC characterized by a relative “cold” immune reactivity.

### 4.2. Histone Modifications: Implication and Inhibition

A growing body of evidence has unequivocally shed light on the relevance of histone-based epigenetic mechanisms in cancer, laying the foundation for a pharmacological approach, in addition to the epigenetic-based strategy (Figure 1).

In pancreatic cancer, different studies provided insights on the crosstalk between the EMT transcription factors ZEB1 and Snail and the HDAC-mediated silencing of E-cadherin expression, contributing to an exacerbation of malignant phenotypes [101,102]. 

A similar mechanism can be found in ovarian cancer, where HDAC3 epigenetically silences the expression of E-cadherin, stimulating cell migration and invasion, while HDAC1 and -2 are associated with increased cell proliferation and to platinum resistance [103,104,105]. Importantly, HDAC1 levels increase over time following cell transformation in ovarian cancer, from an almost undetectable expression to a frank induction in malignant lesions, which is also associated with a poor prognosis, suggesting a role of HDAC1 as a diagnostic biomarker [65,106,107]. 

In line with these observations, in GBM, HDAC1 and -3 levels are positively correlated with WHO tumor grades, reaching the highest expression in recurrent GBM samples. Indeed, HDAC3 expression segregates two populations of GBM patients in which high expression level correlated with the worst prognosis when considered overall survival curve [108,109].

It is not surprising that after the seminal discovery of the HDAC inhibitory activity of sodium butyrate coupled with the breakthrough insights in the epigenetics posed the rationale for HDAC targeting in the clinical setting.

The pharmacological target of the HDAC inhibitors (HDACi) is the zinc ion of the catalytic site of HDAC Classes I, II, and IV; thus, HDAC Class III, utilizing NAD^+^, is intrinsically resistant to these agents (Figure 1). There are four classes of HDACi distinguished for their chemical structure and their specific activity: short-chain aliphatic acid, hydroxymic acids, cyclic tetrapeptide, and benzamides [32]. 

Butyrate, phenylbutyrate, and valproic acid belong to the first family and due to their relatively weak activity, and their use for psychiatric diseases are less clinically interesting [110]. The pan-HDAC inhibitors, hydroxymic acids vorinostat, belinostat, and panobinostat, were initially approved by the FDA in early 2000s for the treatment of myelodysplastic syndrome [32]. The HDAC I-specific inhibitor romidepsin, a cyclic tetrapeptide, has been FDA-approved for T-cell lymphoma [32]. The newest benzamide-based etinostat, a specific class I HDAC inhibitor, is still not FDA-approved, but showed encouraging results in metastatic luminal breast cancer, progressed after anti-estrogen therapy, in combination with examestane [111]. 

Despite the clinical activity that HDACi showed in hematological malignancies, it is glaring to highlight that both the pan- and selective-HDACis, due to the intrinsic pleiotropic nature of HDAC enzymes, exert effects spanning across several hallmarks of cancer, confirmed by the expression alteration of approximately 20% of the genes. Thus, their activity is compatible with several off-target effects and a low safety profile. The main side effects are fatigue, nausea, anorexia, diarrhea, thrombus formation, thrombocytopenia, neutropenia, anemia, myalgia, hypokalemia, and hypophosphatemia. Indeed, the design of a novel class of HDACis to selectively target the critical HDAC isoforms will be an expanding field of pharmaceutical chemistry and may greatly improve the clinical efficacy and attenuate the typical class-effect toxicities. In this field, in a Phase-II trial, platinum-resistant HGSOC patients treated with belinostat experienced severe adverse events without clinical benefits, leading to the termination of the study [112]. Accordingly, in a Phase-I trial, when vorinostat was administered in combination with gemcitabine or carboplatin, it provoked severe hematological toxicities, with a relative partial benefit, not sufficient to proceed the study [113]. 

Similar results were obtain also in GBM; even if HDACi initially showed encouraging activity in preclinical settings, they failed to reach efficacy in clinical trials either as a single agent or in combination with standard therapy temozolomide or targeted-agents bevacizumab [114,115,116]. 

Synergizing combination therapy is represented by the combined administration of 5-azacytidine (DNMTi) and Class I HDACi ITF2357 (givinostat) in HGSOC. In this setting, HGSOC could be sensitized to the immune checkpoint blockade by the upregulation of immune-reactive tumor antigens and subsequent activation of interferon-mediated response [117].

Overexpression of the EHMT2 gene (encoding for the G9a histone methyltransferase, which catalyzes the mono- and bi-methylation of H3K9 and H3K27) is observed in HGSOC and correlated with poor outcome [118].

HGSOC samples are also characterized by the presence of a “poised signature”, typical of embryonic stem cells, which leads to the silencing of genes belonging to the PI3K and TGFβ pathways. This signature is associated with ovarian cancer stem-like proprieties and platinum resistance since it contains both repressive (e.g., H3K27me3) and active (e.g., H3K4me3) marks, commonly encountered in embryonic stem cells [119].

EHMT2 expression correlates with a poor prognosis also in pancreatic cancer. EHMT2 through the H3K9 mark is able to negatively regulate the transcription of Beclin-1, an oncosuppressor gene regulating autophagy. The use of BIX-01294, an EHMT2 inhibitor, is sufficient to rescue Beclin-1 expression, and the combined association with 5-aza-2’-deoxycytidine, a DNMT1 inhibitor, exerts a synergistic effect. This finding supports the rationale for combined inhibition of EHMT2 and DNMT to activate autophagy and arrest cancer progression [120]. 

Functional or pharmacological inhibition of EHMT2 with either shRNA or BIX-01294 strongly activates the autophagy machinery, which in turn induces the differentiation of GBM stem cells [121].

EHMT2 has been demonstrated to regulate also the expression of the CDK inhibitor p27^KIP1^ via the H3K9 mark, leading to an enhanced response to PI3K/mTOR inhibitors in pancreatic cancer cells resistant to gemcitabine [122].

Remarkably, the histone methyltransferase EZH2, a catalytic subunit of the polycomb-repressive complex 2 (PRC2) responsible for the deposit of methyl mark on H3K27, has been reported as a fundamental and necessary contributor to the expansion of CSC compartment and the establishment of drug resistance in HGSOC, as well as PDAC and GBM [123,124,125]. Inhibition of PRC2 in high-grade serous and clear-cell ovarian cancer has been reported to significantly sensitizes cancer cells to platinum-based chemotherapy, widening new clinical avenues for a dose-escalation strategy and improved management of ovarian cancer patients [126,127].

Similar results were achieved in PDAC, where increased nuclear EZH2 expression is correlated with a poor outcome and shorter overall survival. The study also showed that the inhibition of EZH2 in pancreatic tumors induces an upregulation of p27^kip1^, arresting cell cycle progression and strongly sensitizing PDAC cells to gemcitabine treatment [128]. Independent studies concluded that EZH2 indirectly inhibits E-cadherin and RUNX3 (Runx-related transcription factor 3) expression, contributing to the development of malignant phenotypes [129,130]

In GBM, EZH2 acts as an enhancer of proliferation and invasion, finally fostering the expansion of stem compartments [131]. Chen and colleagues mechanistically defined a new axis in which EGFR pathway activation upregulates NEAT1 expression, which in turn triggers β-catenin nuclear localization and mediates H3K9 trimethylation by binding EZH2, resulting in increased cell proliferation [132].

This unexpected clinical failure prompted the development of new epi-drugs targeting HMT (i.e., EZH2) or epigenetic “Readers”, such as BET proteins, which are composed of BRD2, BRD3, BRD4, and BRDT and contain bromodomains that recognize acetylated lysine residues on histone tails.

An emerging application is testified in pediatric glioma, in which H3K27 mutation has been shown to reduce PRC2 activity [133]. Currently, two clinical trials are testing the efficacy of tazemetostat (EZH2 inhibitor) in pediatric glioma with rare gain-of-function mutation of EZH2 or loss-of-function mutation of SMARCB1 or SMARCA4, chromatin remodeling subunits.

To date, contrasting results were achieved in the preclinical application of tazemetostat; while it seemed to be completely ineffective in an in vitro model of H3.3 mutated glioma cells, it reduced the proliferation of glioma cells with H3 or IDH1/2 wild-type background; however, when the same cells were treated for prolonged period, it induced a shift in cell commitment, accelerating tumor progression through an induction of cell proliferation and DNA damage repair [133,134,135,136]. Moreover, it also displayed efficacy in H3K27M-positive glioma and CDKN2A wild-type cells.

Other EZH2 inhibitors, such as CPI-1205, UNC1999 and GSK126, frankly reduced EZH2 activity in PDAC by reducing H3K27 tri-methylation marks, with a biological readout of a slowed proliferation rate and increased apoptosis. In ovarian clear-cell carcinoma, EZH2 inhibition achieved interesting results, and the synthetic lethality was excellent in ARID1A mutated tumors [126]. However, broadened studies are needed to successfully progress into the clinical testing, both in HGSOC and PDAC. 

BET protein family of chromatin readers have come to light recently as suitable therapeutic targets, with excellent results in hematological malignancies [137]. In several studies, BET inhibitors (BETis) has shown to impinge on the expression of MYC oncogene, a master regulator of cell survival and proliferation, as well as one of the most dysregulated genes in cancer [138].

BET proteins work as mediator complex to promote the transcription of several oncogenes, including MYC, through the recognition of and binding to acetyl marks on H3K9 and H3K27. 

In this scenario, the BRD4 gene is frequently overexpressed in PDAC, HGSOC, and GBM, and is deeply associated with survival advantages, resistance to apoptosis, escaping immune-therapy, and overall correlates with a poor prognosis [25,137,139].

Similar to other histone modification inhibitors, BETis dampen cell-cycle progression and tumor progression with improved efficacy and lowered adverse effects. For example, preclinical studies in orthotopic xenograft models demonstrated the high-susceptibility of GBM cells to BETi, such as JQ1, I-BET151, GS-626510, and OTX015, although the last one did not improve the clinical outcome of GBM patients [140,141,142].

Interestingly, BETi could act as an enhancer in combination with other targeted therapies. In breast cancer and HGSOC, the combination of BETi plus PARP inhibitor resulted in a pronounced induction of DNA damage, likely due to a lack in the DNA homologous recombination [143]. A phase Ib clinical trial is now ongoing and aims to test the combination of anti-PD-L1 immune checkpoint therapy (atezolizumab) plus RO6870810 BETi in patients with HGSOC or triple-negative breast cancer NCT03292172 (Table 1).

## 5. Conclusions

Historically, since the dawn of pathology, outstanding scientists have recognized and classified the frank similitudes in the histological organization and morphology of cancer cells, intrinsically resembling features typical of embryonic stem cells.

The genomic era meticulously reported a plethora of profiled mutations across different tumor types, although the molecular reductionist description of cancer, and generally of cell phenotypes, as results of simplistic genetic expression is no longer accepted.

The post-genomic era and the relative novel epigenomic revolution simply uncovered a Pandora’s box of cell biology, fueling the complexity beyond the myriad of cell states in developmental systems and cancers. 

The unparalleled efforts and the applied technologies depicted a complex network in the heart of cell biology, unraveling the traditional idea that cell phenotype lies in the DNA sequence. 

The multiple observations, in many cancer types, of aberrant DNA methylation within the promoter of several oncosuppressor genes, the “poised histone marks” of cancer stem cells, along with the high percentage of mutations in epigenetic regulators posed the epigenomic machinery at the balance between development and de-differentiation. A huge body of evidence clearly linked epigenetic dysregulation to tumorigenesis and cancer progression. The thin conductive thread underpinning the plasticity nature of cancer stem cells is still undisclosed, and we need to refine the biological system to follow-up the mechanism by which CSCs adapt and reshape their repertoire of gene expression in dynamic and multidimensional modalities, in a time- and tissue-specific fashion.

To date, we still lack the ability to fill the molecular gap between the rude observation of similarities between embryonic stem cells and cancer stem cells at epigenetic layers and gene expression. Is it possible that within a cell there are hidden epigenetic vulnerabilities, which, in turn, may predispose to cancer development? Which are the epigenetic mechanisms that are turned-on during tumorigenesis? How does the epigenetic machinery control cell hierarchy? Or symmetric versus asymmetric cell division? Is it possible that epigenetic modulators segregate differentially across daughter cells? If yes, which is the signal inducing the segregation? And finally, is there the possibility to functionally exhaust the cancer stem compartment through epigenetic reprogramming? 

The next challenge will certainly be the understanding of epigenetic control of cell fate. A step back should be the understanding of the different epigenetic regulatory pathways in adult and embryonic stem cells; in this context, cancer is considered to be an age-related disease in a stochastic model. 

In other words, we should study the role of epigenetic modulators in cell physiology beyond the chromatin remodeling action. Considering that many epigenetic modulators are ubiquitously distributed into the cells and the redundant nature of the landscape of genes involved in the control of chromatin architecture, it would be conceivable to hypothesize at least a double role: the canonical epigenetic regulation and the non-canonical function.

At a glance, the intrinsic tumor heterogeneity and the underlying epigenetic dependencies offer an arsenal of potential pharmacological avenues to target undifferentiated tumors.

However, the testing of new “epidrugs” raised a discrepancy between the preclinical setting and the clinical benefits. In several in vitro and in vivo experiments, the vast majority of epigenetic inhibitors exerted excellent results in the control of tumor growth, progression, eventually sensitizing tumor cells to combined chemotherapy or immunotherapy. Despite these encouraging results, when translated into the clinic, different compounds showed modest activity against solid tumors. Indeed, the exploited models cannot be considered as surrogates of tumor complexity. The main critical aspects reside at biochemical, pharmacological, and technological levels. 

Overall, many steps forward have been taken in epigenetic studies, but several different issues must be taken into account in order to practically translate new “from the bench to the bedside” pharmacological tools involving epigenetic regulation. Finally, failures in the application of the “epidrugs” in a clinical context need to be a starting point for prompting research on cancer epigenetics, clarifying the blind spots and unmeant connections of this fascinating side of cancer biology.

## Figures and Tables

**Figure 1 cancers-12-00682-f001:**
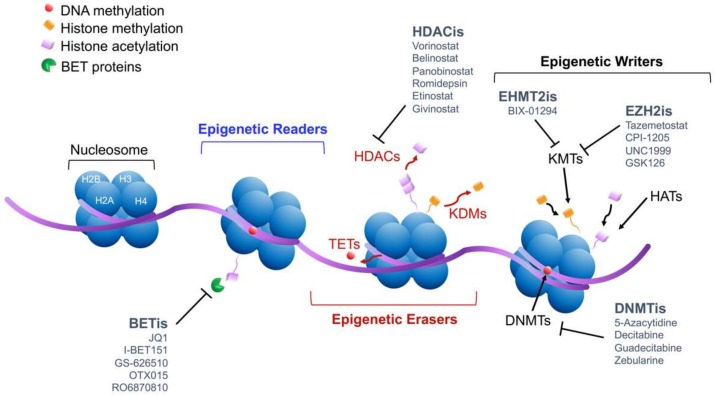
Mechanisms of epigenetic regulation and pharmacological inhibition. Schematic representation of nucleosome, the elementary unit of chromatin, and the main mechanisms of chromatin modifications, including DNA methylation and histone modifications. In grey, typical epigenetic inhibitors tested in clinical or preclinical settings.

**Table 1 cancers-12-00682-t001:** Summary of epigenetic inhibitors in preclinical studies or clinical trials.

**Class**	**Compound**	**Target**	**Clinical status**
**DNMTi**	5-Azacytidine	pan-DNMT	Approved: AML, MDS (EMA and FDA)
	Decitabine	pan-DNMT	Approved: AML (EMA), MDS (FDA)
	Guadecitabine	pan-DNMT	Clinical Trials: MDS, AML, OC, HCC
	Zebularine	pan-DNMT	Preclinical
**HDACi**	**Hydroxymic acids**		
	Vorinostat	Class	Approved: CTCL (FDA)
	Belinostat	Classes I, II	Approved: PTCL (EMA and FDA)
	Panobinostat	Classes I, II, IV	Approved: MM (EMA and FDA)
	Givinostat	Classes I, II	Clinical trials: MM, CMD
	**Cyclic peptide derivates**	**Target**	**Clinical status**
	Romidepsin	Class I	Approved: CTCL, PTCL (EMA and FDA)
	**Benzamide derivates**	**Target**	**Clinical status**
	Entinostat	Class I	Clinical trials: NSCLC, BC, CNS tumors, PAC, OC, RCC, PDAC
	Tacedinaline	Class I	Clinical trials: MM, NSCLC, PDAC
	**Fatty acid derivates**	**Target**	**Clinical status**
	Phenylbutyrate	Classes I, II	Clinical trials: CNS tumors, AML, MM, MDS, MPS, NSCLC, PAC
	Valproic acid	Classes I, II	Clinical trials: AML, MDS, HNSCC, CNS tumors, advanced cancers
**KMTi**	Tazemetostat (EPZ-6438)	EZH2	Clinical trials: solid tumors, DLBCL, HL, non-HL, MRT, advanced solid tumors
	CPI-1205	EZH2	Clinical Trials: BCL, advanced solid tumors, mCRPC
	GSK126 (or GSK2816126)	EZH2	Clinical trials: DLBCL, NHL, MM
	UNC1999	EZH2	Preclinical
	BIX-01294	G9A (EHMT2)	Preclinical
**BETi**	JQ1	pan-BET	Preclinical
	I-BET151	pan-BET	Preclinical
	GS-626510	pan-BET	Preclinical
	OTX015	pan-BET	Clinical trials: AML, DLBCL, ALL, MM, GBM, NMC, BC, NSCLC, CRCP
	RO6870810	pan-BET	Clinical trials: AML, MDS, advanced solid tumors, DLBCL

ALL: acute lymphoblastic leukemia, AML: acute myeloid leukemia, BC: breast cancer, BCL: B-cell lymphoma, CMD: chronic myeloproliferative diseases, CNS: central nervous system, CRPC: castration-resistant prostate cancer, CTCL: cutaneous T-cell lymphoma, DLBCL, diffuse large B-cell lymphoma, HCC: hepatocellular carcinoma, HNSCC: head and neck squamous cell carcinoma, mCRPC: metastatic castration-resistant prostate cancer, MDS: Myelo–Dysplastic syndrome, MLL: myeloid lymphoblastic leukemia, MM: multiple myeloma, MRT: malignant rhabdoid tumors, NHL: non-Hodgkin lymphoma, NSCLC: non-small cell lung cancer, NMC: NUT midline carcinoma, OC: ovarian cancer, PAC: prostate adenocarcinoma, PDAC: pancreatic ductal adenocarcinoma, PTCL: perypheral T-cell lymphoma, RCC: renal cell carcinoma.

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
