# Peer review of "Targeting Epigenetic Dependencies in Solid Tumors: Evolutionary Landscape Beyond Germ Layers Origin"

_cancers, 2020, doi:10.3390/cancers12030682_

Round 1

Reviewer 1 Report

The paper of Francesca Citron et al. “Targeting epigenetic dependencies in solid tumors: evolutionary landscape beyond germ layers origin” give an overview of the principal epigenetic dependencies of solid tumors, from the molecular mechanisms to the pharmacological approach targeting this pathway.

The Authors try to specifically focus on the tumors arising from the three different germ layers: glioblastoma multiforme (GBM), high grade epithelial  ovarian cancer (HGEOC), pancreatic ductal adenocarcinoma (PDAC).

I considered that this review was very well and clear done. Moreover, I think that overall this manuscript can be very helpful to understand about molecular mechanisms of epigenetic modifications. 

Author Response

I considered that this review was very well and clear done. Moreover, I think that overall this manuscript can be very helpful to understand about molecular mechanisms of epigenetic modifications.

            We would like to thank the reviewer for the positive comment

Reviewer 2 Report

Here are some of the comments:

In the introduction "Starting from the embryogenesis, the DNA and chromatin are the responsible for the hierarchical 60 development of any human tissue, driving the transition from the multipotent stem and progenitor 61 cells to terminally differentiated and specialized progeny (2, 8, 9)" requires rephrasing/correction as too many 'the' are added.

Next,  "The dynamic of gene expression patterns' could be replaced by "Dynamics of gene expression pattern".

In subheading 2, 'transcriptionally repress state' could be replaced by 'repressed' .

It is unclear what "On this base" means?

To correct the sentence in subheading 3--'Moreover, dedifferentiate cells display a dynamic'-- use 'dedifferentiated cell'.

This sentence needs to be rephrased "The standard of care for PDAC patients is represented by cytotoxic chemotherapy, primarily FOLFIRINOX and gemcitabine-based regimens, but diagnosis in advanced or metastatic stages pose a clinical challenge, and there no effective therapeutics to achieve long-term remissions of the disease (70)."

Author Response

In the introduction "Starting from the embryogenesis, the DNA and chromatin are the responsible for the hierarchical 60 development of any human tissue, driving the transition from the multipotent stem and progenitor 61 cells to terminally differentiated and specialized progeny (2, 8, 9)" requires rephrasing/correction as too many 'the' are added.

            We would like to thank the review for the helpful comments and suggestions. We agree with the reviewer and we modified this sentence accordingly.

-"The dynamic of gene expression patterns' could be replaced by "Dynamics of gene expression pattern".

-In subheading 2, 'transcriptionally repress state' could be replaced by 'repressed' .

-It is unclear what "On this base" means?

-To correct the sentence in subheading 3--'Moreover, dedifferentiate cells display a dynamic'-- use 'dedifferentiated cell'.

-This sentence needs to be rephrased "The standard of care for PDAC patients is represented by cytotoxic chemotherapy, primarily FOLFIRINOX and gemcitabine-based regimens, but diagnosis in advanced or metastatic stages pose a clinical challenge, and there no effective therapeutics to achieve long-term remissions of the disease (70)."

            We would like to thank again the reviewer for the suggestions, we modified the text as requested. We also substituted “on this base” with “indeed” at line 116 and we rephrased the sentence at line 230.

Reviewer 3 Report

This review entitled « targeting epigenetic dependencies in solid tumors: evolutionary landscape beyond germ layers origin” is written by Francesca Citron & Linda Fabris. They summarized the role of epigenetics in cancer and mainly epigenetic modifications which may early affect cancer cells and promote tumorigenesis.

Although many reviews previously focused on a similar topic, this present review remains interesting and could be considered for publication in “Cancers” after few modifications.

-line 263-275. Percentages of tumors concerned by these modifications should be given (as done for RASSF1a line 276)

-line 293 “these findings support a strong rationale for the clinical application of… DNMTi” ; This is not as simple as that. For example, authors mentioned later that methylation of MGMT is a good biomarker for TMZ response in glioma. Molecular side-effects of DNMTi could be better discussed. For example, see (Lamparska and Smith, Epigenetics Diagnosis & Therapy 2015) for effects of subproducts of DNMTi. See also, (Pacaud R et al Scientific reports 2014) (Gaudet F et al Science 2003) showing that inhibition of DNMT or invalidation of DNMT promotes tumorigenesis.

-line 311: not clear the affirmation concerning the activation of TGFb pathway as a good prognosis. What about EMT signaling? This point could be developed.

-line 322-324: not clear how “MGMT mutated” could be overexpressed following DNMTi treatment and counteract TMZ efficacy. Why mutated?

-line 330-335. Example concerning the HDACi-induced NYESO1 expression is interesting but should be better explained at the molecular level. For example, (Cartron PF et al Molecular oncology 2013) dissected the epigenetic regulation of this gene by cooperation between DNMTs/G9a/HDAC.

-line 361: concerning the role of autophagy activation following EZH2i treatment and its positive effect against cancer cells. May be in this paragraph, it could be also reminded that both pro and anti-cancer properties of autophagy have been described. Indeed, anticancer effects of BIX is may be more complicated that a simple induction of autophagy. For example, see reviews (liu L et al 2020, cancers), (Kwon Y et al Cancers 2019), (Liu L et al Oncotarget 2017)

-Line 394: The paragraph concerning “HDACi” could be moved following the paragraph concerning “HDAC”. The paragraph on EZH2 could be associated with the EZH2i. I guess it would be easier to read.

-line 427 “new class HDACi targeting EZH2”. Not clear, EZH2 is not a HDAC but a HMT ? line 455, a similar sentence is not clear “other histone modification inhibitor BETi…” bromodomain-containing proteins are not modifiers of histones but readers.

-in conclusion, authors mentioned that clinical assays using DNMTi frequently failed. I guess that a paragraph could be added, concerning the specific recruitment of DNMT by transcriptional factors. A target of DNMT/TF complexes could be more specific and less toxic. For example, see (Wang YA et al, cancer Biol ther 2005),  (Senyuk V et al PlosOne 2011), (Cheray M et al Theranostics 2016), (Hervouet E et al Epigenetics 2009)…

-line102: Histone acetyl transferase abbreviation is HAT and not HAC

-some typo mistakes are frequently found in the text. Please, read carefully the paper and edit it.v

Author Response

This review entitled « targeting epigenetic dependencies in solid tumors: evolutionary landscape beyond germ layers origin” is written by Francesca Citron & Linda Fabris. They summarized the role of epigenetics in cancer and mainly epigenetic modifications which may early affect cancer cells and promote tumorigenesis. Although many reviews previously focused on a similar topic, this present review remains interesting and could be considered for publication in “Cancers” after few modifications

            We would like to thank the reviewer for the positive comment regarding our work.

line 263-275. Percentages of tumors concerned by these modifications should be given (as done for RASSF1a line 276).

            We agree with the reviewer, so we added the percentages of mutation of the modifications cited, line 278.

-These findings support a strong rationale for the clinical application of… DNMTi” ; This is not as simple as that. For example, authors mentioned later that methylation of MGMT is a good biomarker for TMZ response in glioma. Molecular side-effects of DNMTi could be better discussed. For example, see (Lamparska and Smith, Epigenetics Diagnosis & Therapy 2015) for effects of subproducts of DNMTi. See also, (Pacaud R et al Scientific reports 2014) (Gaudet F et al Science 2003) showing that inhibition of DNMT or invalidation of DNMT promotes tumorigenesis.

            We discussed about side effects of DNMTs inhibitors, as they are common between most of the alkylating agents. We cited the article by Pacaud R et al, and we would like to thank the reviewer for the helpful suggestion.

-line 311: not clear the affirmation concerning the activation of TGFb pathway as a good prognosis. What about EMT signaling? This point could be developed

            We reformulated our sentence to make it clearer for the reader. We didn’t include a paragraph regarding EMT, since in our opinion it is out of topic for this review.

-line 322-324: not clear how “MGMT mutated” could be overexpressed following DNMTi treatment and counteract TMZ efficacy. Why mutated?

            We thank the reviewer, MGMT is indeed methylated, not mutated.

-line 330-335. Example concerning the HDACi-induced NYESO1 expression is interesting but should be better explained at the molecular level. For example, (Cartron PF et al Molecular oncology 2013) dissected the epigenetic regulation of this gene by cooperation between DNMTs/G9a/HDAC

            We included a section discussing this point, and the relative reference, as suggested by the reviewer, lines 329-342.

-line 361: concerning the role of autophagy activation following EZH2i treatment and its positive effect against cancer cells. May be in this paragraph, it could be also reminded that both pro and anti-cancer properties of autophagy have been described. Indeed, anticancer effects of BIX is may be more complicated that a simple induction of autophagy. For example, see reviews (liu L et al 2020, cancers), (Kwon Y et al Cancers 2019), (Liu L et al Oncotarget 2017)

            We thank the author for the observation, even if we think that focusing of autophagy mechanism would outside our focus in this review. Nonetheless, we agree that this effect would require more discussion, in a more molecular-oriented review.

-Line 394: The paragraph concerning “HDACi” could be moved following the paragraph concerning “HDAC”. The paragraph on EZH2 could be associated with the EZH2i. I guess it would be easier to read.

-line 427 “new class HDACi targeting EZH2”. Not clear, EZH2 is not a HDAC but a HMT ? line 455, a similar sentence is not clear “other histone modification inhibitor BETi…” bromodomain-containing proteins are not modifiers of histones but readers.

- line102: Histone acetyl transferase abbreviation is HAT and not HAC

We agree with the reviewer and we modified all points accordingly. We also moved the paragraph regarding HDACi and EZH2i, as the reviewer suggested

-in conclusion, authors mentioned that clinical assays using DNMTi frequently failed. I guess that a paragraph could be added, concerning the specific recruitment of DNMT by transcriptional factors. A target of DNMT/TF complexes could be more specific and less toxic. For example, see (Wang YA et al, cancer Biol ther 2005), (Senyuk V et al PlosOne 2011), (Cheray M et al Theranostics 2016), (Hervouet E et al Epigenetics 2009)

            We carefully revised the literature concerning the possibility to target the epigenetic machinery in three different cancer types. We strongly agree with Reviewer #3, since the use of DNMTi has demonstrated to be a failing strategy, due to their low specificity and the consequent plethora of side effects.

However, the combined targeting of the DNMTs complexed with a specific TF has not been applied to clinic so far, and, for our knowledge, only one research article demonstrated the efficacy of this combination in a subcutaneous model of GBM (nude mice).

For this reason, we are convinced that these data should be independently confirmed by other groups and possibly in stronger preclinical settings. Indeed, for the purpose of this review, we did not include a paragraph concerning this undoubtedly new and promising approach.